# Structural basis for anion conduction in the calcium-activated chloride channel TMEM16A

**Cristina Paulino, Yvonne Neldner, Andy KM Lam, Valeria Kalienkova, Janine Denise Brunner[†], Stephan Schenck[‡], Raimund Dutzler***

Department of Biochemistry, University of Zurich, Zurich, Switzerland

**Abstract** The calcium-activated chloride channel TMEM16A is a member of a conserved protein family that comprises ion channels and lipid scramblases. Although the structure of the scramblase nhTMEM16 has defined the architecture of the family, it was unknown how a channel has adapted to cope with its distinct functional properties. Here we have addressed this question by the structure determination of mouse TMEM16A by cryo-electron microscopy and a complementary functional characterization. The protein shows a similar organization to nhTMEM16, except for changes at the site of catalysis. There, the conformation of transmembrane helices constituting a membrane-spanning furrow that provides a path for lipids in scramblases has changed to form an enclosed aqueous pore that is largely shielded from the membrane. Our study thus reveals the structural basis of anion conduction in a TMEM16 channel and it defines the foundation for the diverse functional behavior in the TMEM16 family.

*For correspondence: dutzler@bioc.uzh.ch

Present address: [†]Department Biozentrum, University of Basel, Basel, Switzerland; [‡]Laboratory of Biomolecular Research, Paul Scherrer Institute, Villigen, Switzerland

Competing interests: The authors declare that no competing interests exist.

## Introduction

Calcium-activated chloride channels (CaCCs) are important constituents of diverse physiological processes, ranging from epithelial chloride secretion to the control of electrical excitability in smooth muscles and neurons (*Hartzell et al., 2005*; *Huang et al., 2012*; *Oh and Jung, 2016*; *Pedemonte and Galietta, 2014*). These ligand-gated ion channels are activated upon an increase of the intracellular $Ca^{2+}$ concentration as a consequence of cellular signaling events. Although CaCC function can be accomplished by unrelated protein architectures (*Kane Dickson et al., 2014*, *Kunzelmann et al., 2009*), the so far best-characterized processes are mediated by the protein TMEM16A (*Caputo et al., 2008*; *Schroeder et al., 2008*; *Yang et al., 2008*). TMEM16A is a member of the large TMEM16 family of membrane proteins, also known as anoctamins (*Yang et al., 2008*). The family is exclusively found in eukaryotes and contains 10 paralogs in mammals that all share considerable sequence homology (*Milenkovic et al., 2010*) (*Figure 1—figure supplement 1*). Although it was initially anticipated that all TMEM16 proteins would function as anion channels (*Hartzell et al., 2009*; *Tian et al., 2012*; *Yang et al., 2008*), it is now generally accepted that only two family members (the closely related TMEM16A and B) are ion channels (*Pifferi et al., 2009*; *Scudieri et al., 2012*), whereas most others work as lipid scramblases, which catalyze the passive and bidirectional diffusion of lipids between the two leaflets of a phospholipid bilayer (*Brunner et al., 2016*; *Malvezzi et al., 2013*; *Suzuki et al., 2013*, *2010*; *Whitlock and Hartzell*, *2017*, *2016*).

The TMEM16 family shows a new protein fold, as revealed by the structure of the fungal homologue nhTMEM16, which functions as lipid scramblase (*Brunner et al., 2014*). nhTMEM16 consists of structured cytoplasmic N- and C-terminal components and a transmembrane domain (TMD) containing 10 transmembrane helices. As general for the TMEM16 family, the protein is a homo-dimer

**eLife digest** Cell membranes are made up of two layers of oily molecules, called lipids, embedded with a variety of proteins. Each type of membrane protein carries out a particular activity for the cell, and many are involved in transporting other molecules from one side of the membrane to the other.

The TMEM16 proteins are a large family of membrane proteins. Most are known as lipid scramblases and move lipids between the two layers of the membrane. However, some TMEM16 proteins transport ions in or out of the cell, and are instead called ion channels. TMEM16 proteins are found in animals, plants and fungi but not bacteria, and play key roles in many biological activities that keep these organisms alive. For example, in humans, ion channels belonging to the TMEM16 family help keep the lining of the lung moist, and allow muscles in the gut to contract.

The structure of a scramblase shows that two protein units interact, with each unit containing a furrow that spans the membrane, through which lipids can move from one layer to the other. However, to date, the shape of a TMEM16 ion channel has not been determined. It was therefore not clear how a protein with features that let it transport large, oily molecules like lipids had evolved to transport small, charged particles instead.

TMEM16A is a member of the TMEM16 family that transports negatively charged chloride ions. Using a technique called cryo-electron microscopy, Paulino et al. have determined the three-dimensional shape of the version of TMEM16A from a mouse. Overall, TMEM16A is organized similarly to the lipid scramblase. However, some parts of the TMEM16A protein have undergone rearrangements such that the membrane-exposed furrow that provides a path for lipids in scramblases is now partially sealed in TMEM16A. This results in an enclosed pore that is largely shielded from the oily membrane and through which ions can pass. Additionally, biochemical analysis suggests that TMEM16A forms a narrow pore that may widen towards the side facing the inside of the cell, though further work is needed to understand if this is relevant to the protein's activity.

The three-dimensional structure of TMEM16A reveals how the protein's architecture differs from other family members working as lipid scramblases. It also gives insight into how TMEM16 proteins might work as ion channels. These findings can now form a strong basis for future studies into the activity of TMEM16 proteins.

(*Fallah et al., 2011*; *Sheridan et al., 2011*) with each subunit containing its own lipid translocation path located at the two opposite corners of a rhombus-shaped protein distant from the dimer inter-face (*Brunner et al., 2014*). This lipid path is formed by the 'subunit cavity', a membrane-spanning furrow of appropriate size to harbor a lipid headgroup. Since the subunit cavity is exposed to the membrane, it was proposed that its polar surface provides a favorable environment for lipid head-groups on their way across the membrane, whereas the fatty-acid chains remain embedded in the hydrophobic core of the bilayer (*Brunner et al., 2014*). In the vicinity of each subunit cavity, within the membrane-embedded domain, a conserved regulatory calcium-binding site controls the activity of the protein (*Brunner et al., 2014*). In light of the nhTMEM16 structure and the strong sequence conservation within the family, a central open question concerns how the TMEM16A architecture has adapted to account for its altered functional properties. Previous results suggested that the same region constituting the scrambling path also forms the ion conduction pore (*Yang et al., 2012*; *Yu et al., 2012*). However, in what way the distinct structural features of a scramblase, which allows the diffusion of a large and amphiphilic substrate, are altered in a channel that facilitates the transmembrane movement of a comparably small and charged anion, remained a matter of controversy.

Here we have resolved this controversy by the structure determination of mouse TMEM16A (mTMEM16A) by cryo-electron microscopy (cryo-EM) at 6.6 Å resolution and a complementary electrophysiological characterization of pore mutants. Our data define the general architecture of a calcium-activated chloride channel of the TMEM16 family and reveal its relationship to the majority of family members working as lipid scramblases. The protein shows a similar overall fold and dimeric organization as the lipid scramblase nhTMEM16. However, conformational rearrangements of helices

lining the lipid scrambling path have sealed the subunit cavity, resulting in the formation of a protein-enclosed ion conduction pore that is for most parts shielded from the membrane but that might be partly accessible to lipids on its intracellular side.

# Results

## Structure determination

We were interested in the structural properties that distinguish ion channels from lipid scramblases in the TMEM16 family and thus decided to investigate the structural properties of the chloride channel TMEM16A by single particle cryo-EM. For that purpose, we generated a stable HEK293 cell-line, which constitutively expresses the (*ac*) isoform of mTMEM16A, and purified the protein at a saturating calcium concentration in the detergent digitonin (*Figure 1—figure supplement 2A,B*). Images of flash-frozen samples were recorded on a FEI TITAN Krios electron microscope equipped with an energy filter and a K2-summit camera (*Figure 1—figure supplement 2C*). The three-dimensional structure of the mammalian ion channel at a nominal resolution of 6.6 Å was reconstructed from total of 213,243 particles picked from 4178 micrographs (*Figure 1—figure supplement 2C,D*; *Figure 1—figure supplement 3A*; and *Table 1*). Since the resolution did not significantly improve after addition of further images, it is likely limited by the sample. In the resulting electron density map, the main features of the protein are well defined (*Figure 1A*, *Figure 1—figure supplement 4* and *Video 1*). Similarities with nhTMEM16 allowed the construction of a poly-alanine model

**Table 1.** Statistics of cryo-EM data collection, 3D reconstruction and model building.

| Data collection | |
| --- | --- |
| Microscope | FEI Titan Krios |
| Voltage (kV) | 300 |
| Camera | Gatan K2-summit |
| Camera mode | super-resolution |
| Energy filter | post-column Gatan GIF quantum energy filter (20 eV slit) |
| Defocus range (μm) | −0.5 to −3.8 |
| Pixel size (Å) | 0.675 (in super-resolution) 1.35 (for reconstruction) |
| Objective aperture (μm) | 100 |
| Exposure time (s) | 15 |
| Number of frames | 50 or 100 |
| Dose rate on specimen level (e⁻/Å²) | 0.8 or 1.5 per frame ~80 in total |
| **Reconstruction** | |
| Software | RELION1.4 and RELION2.0 |
| Symmetry | C2 |
| Final number of refined particles | 213,243 |
| Resolution of polished unmasked map (Å) | 7.85 Å |
| Resolution of polished masked map (Å) | 6.65 Å |
| Map sharpening B-factor (Å²) | −351 (−700 for model building) |
| **Model Statistics** | |
| Number of residues modeled | 434 |
| Software | Chimera, Coot, Phenix |
| Map CC (whole unit cell) | 0.552 |
| Map CC (around assigned model) | 0.857 |

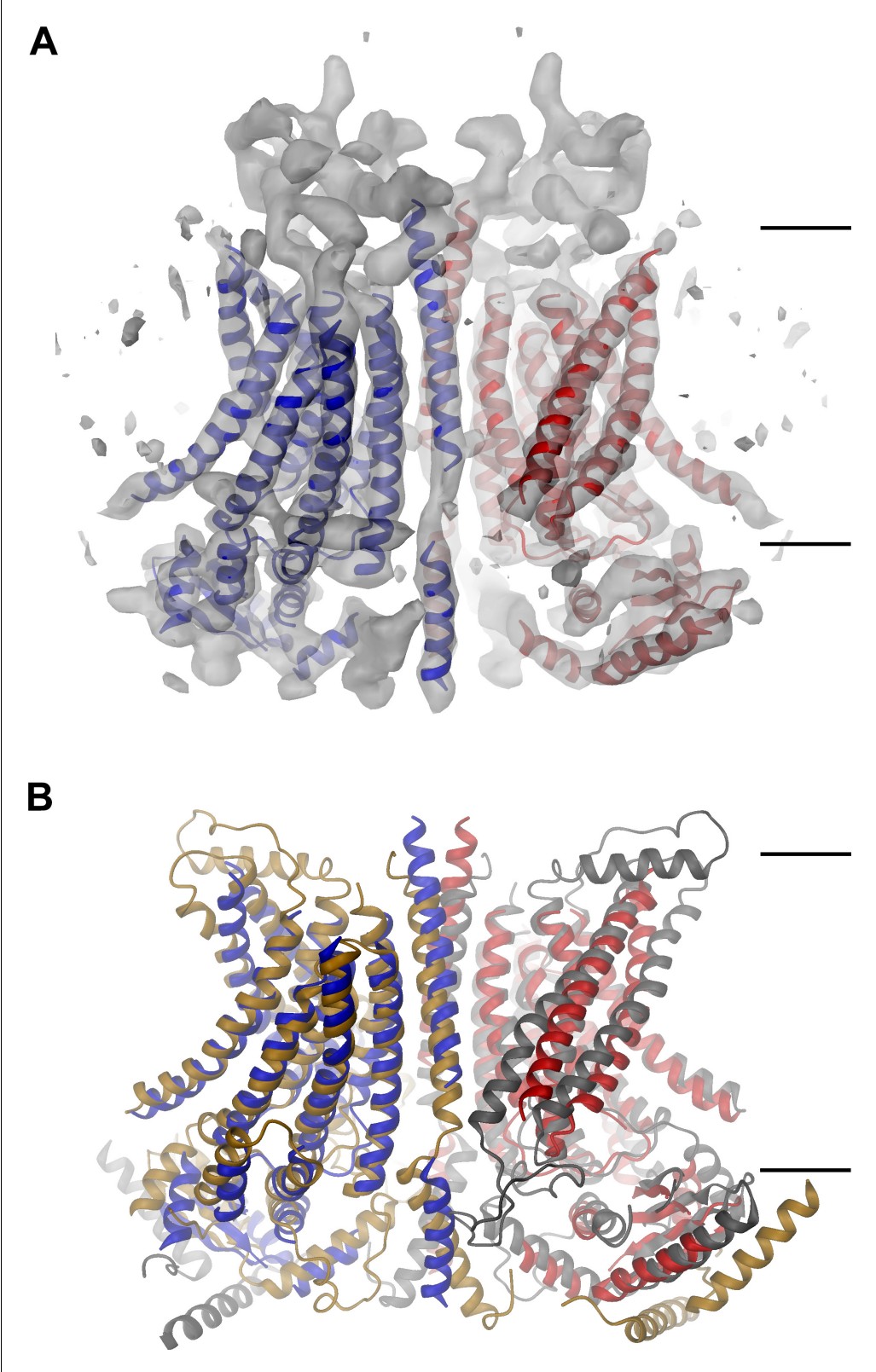

**Figure 1.** mTMEM16A structure. (**A**) Ribbon representation of the mTMEM16A dimer with the EM density (contoured at 11σ) superimposed. (**B**) Superposition of mTMEM16A (blue and red) and nhTMEM16 (beige and grey). A and B, The view is from within the membrane with the extracellular side at the top. The membrane boundary is indicated.

*Figure 1 continued on next page*

*Figure 1 continued*

The following figure supplements are available for figure 1:

**Figure supplement 1.** Sequence alignment.

**Figure supplement 2.** Protein preparation and cryo-EM image processing.

**Figure supplement 3.** Three-dimensional reconstruction of mTMEM16A.

**Figure supplement 4.** EM density of the mTMEM16A channel.

encompassing the secondary structure elements of the TMD and most of the cytoplasmic N- and C-terminal domains (*Figure 1—figure supplement 4A*).

## mTMEM16A structure

The EM-density of mTMEM16A superimposed on the model of the protein is shown in *Figure 1A*. Due to the presence of $Ca^{2+}$, it likely shows the channel in a $Ca^{2+}$-bound conformation. In light of the irreversible rundown of TMEM16A-mediated currents observed in patch-clamp experiments at high $Ca^{2+}$ concentrations, it is at this point ambiguous whether this conformation corresponds to a conducting or a non-conducting state of the channel. Within the membrane, the overall dimensions of mTMEM16A are very similar to nhTMEM16 (*Figure 1B*, *Figure 1—figure supplement 4B* and *Video 1*). All transmembrane helices are well resolved and thus, could be unambiguously allocated. On the extracellular side, the mTMEM16A map contains a substantial amount of unassigned density that can be attributed to extended loops connecting transmembrane α-helices 1–2 (α1α2 loop) and transmembrane α-helices 9–10 (α9α10 loop), which are respectively 50 and 65 residues longer compared to nhTMEM16 (*Figure 1A* and *Figure 1—figure supplement 1*). Both loops appear to be structured, folding into a compact extracellular domain (*Figure 2A*). Notably, this domain harbors six cysteines that have been shown to be indispensable for channel activity (*Yu et al., 2012*) and that are thus potentially involved in disulfide bridges. On the cytoplasmic side, the N-terminal domain of mTMEM16A exhibits a similar fold and location with respect to the TMD as its counterpart in nhTMEM16 (*Figures 1B* and *2B,C* and *Figure 1—figure supplement 4B*). Consequently, there is no interaction between the N-terminal domains of adjacent subunits, which was previously proposed based on biochemical experiments (*Tien et al., 2013*). A 92 residue long extension in mTMEM16A that precedes the folded N-terminal domain (*Figure 1—figure supplement 1*) appears to be unstructured, but there is unaccounted electron density that cannot be interpreted at the current resolution of the data (*Figure 1—figure supplement 4B–D*). At the C-terminus, which is 38 residues shorter than its equivalent part in nhTMEM16, the first α-helix (Cα1) is folded but it has moved away from the dimer axis and thus no longer contacts its symmetry mate (*Figure 2D*). The remainder of the C-terminus is likely unstructured and, unlike in nhTMEM16, does not interact with the adjacent subunit. Hence, the interaction of the subunits within the mTMEM16A dimer differs significantly from nhTMEM16 since the cytosolic domains do not contribute to the dimer interface. Instead, interactions are established mainly at the extracellular part of transmembrane α-helix 10, which

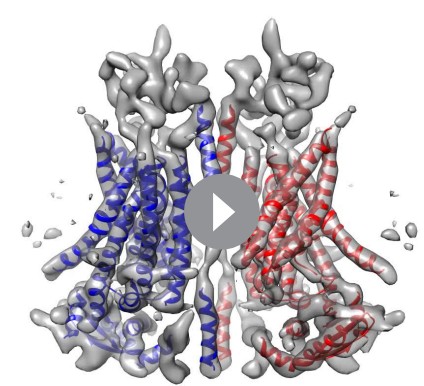

**Video 1.** The mTMEM16A structure. Ribbon representation of the mTMEM16A model with the EM density and the nhTMEM16 structure superimposed. The structures are seen from within the membrane. Ribbons are colored as in *Figure 1* and the positions of bound $Ca^{2+}$ in nhTMEM16 are indicated by green spheres.

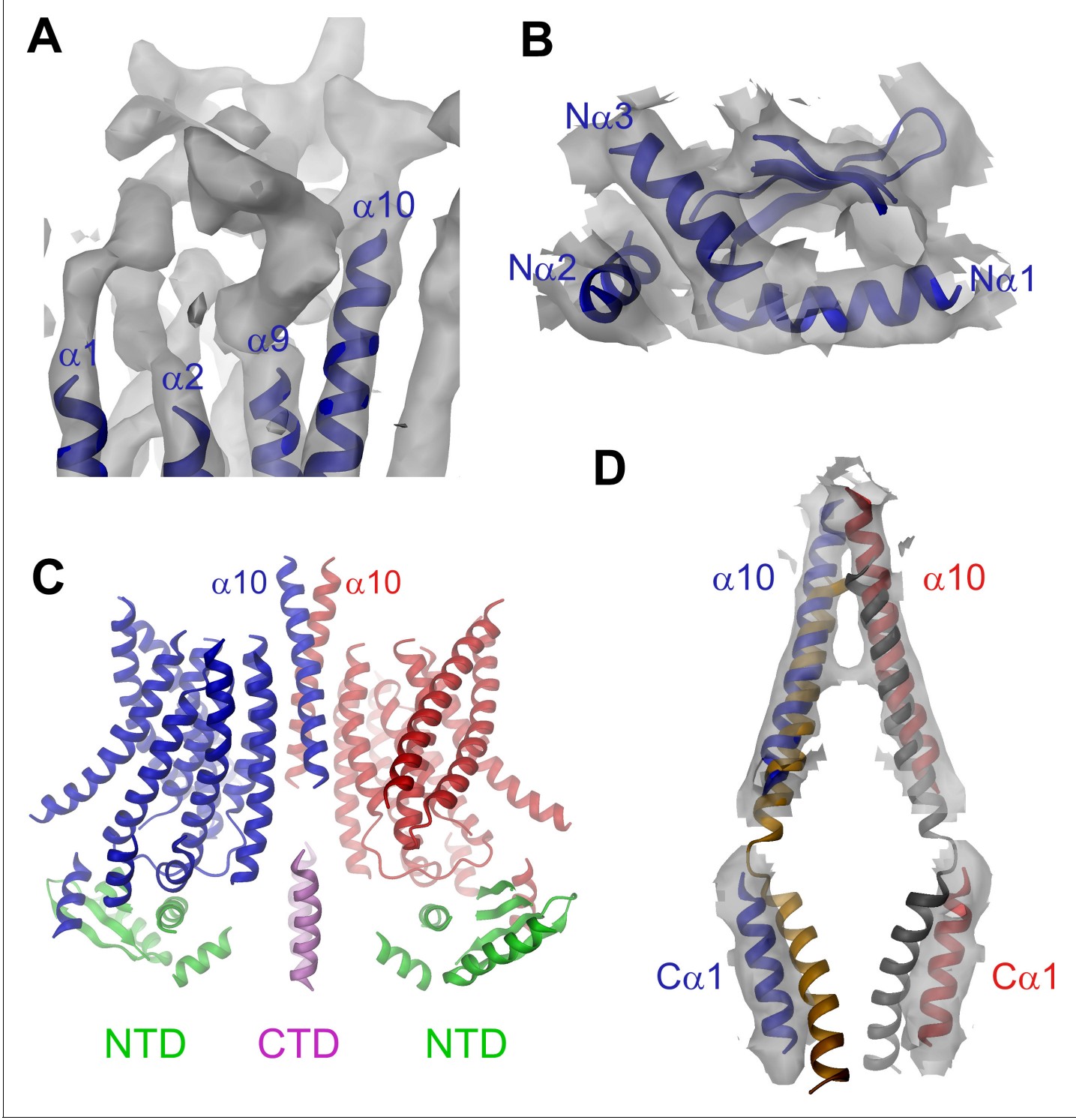

**Figure 2.** Features of the mTMEM16A structure. (**A**) Unassigned EM density (contoured at 11σ) of the extracellular α1α2 and α9α10 loops. Connected transmembrane α-helices are shown as ribbon and labeled. (**B**) Structure of the N-terminal domain. Secondary structure elements are shown as ribbon, α-helices are labeled. (**C**) Ribbon representation of the mTMEM16A dimer. The transmembrane domains (TMD) of individual subunits are colored in blue and red, respectively, N-terminal domains (NTD) in green and the C-terminal domains (CTD) in violet. The view is as in *Figure 1A*. (**D**) Helices α10 of the TMD and Cα1 of the CTD of both subunits of the superimposed dimeric mTMEM16A and nhTMEM16 structures are shown. The view is from within the membrane towards the dimer interface. B and D, Sections of the EM density (contoured at 7σ) are superimposed on selected parts of the model.

is in a similar location as in nhTMEM16 but extends further towards the outside (*Figures 1* and *2D*). In the TMD, all membrane-spanning segments are well defined including two short amphiphilic α-helices at its N-terminal part that interact with the polar headgroups at the inner leaflet of the lipid bilayer (*Figure 1—figure supplement 4E*). In general, the transmembrane helices are in comparable locations to their counterparts in nhTMEM16 (*Figure 1B* and *Figure 1—figure supplement 4B*) and thus account for the overall similarity between both structures.

## The pore region

The pore region of mTMEM16A, also containing the regulatory calcium-binding site, is formed by transmembrane α-helices 3–8. This region is well defined, except for the loops connecting α-helices 5 and 6 and 6 and 6' (*Figure 3*, *Figure 3—figure supplement 1* and *Video 2*). Although, at the current resolution, neither the helix-pitch nor side-chains are resolved, there are several structural features that constrain the location of residues and thus allow for their approximate assignment. The placement is facilitated by conserved loops connecting α-helices 4–5, 7–8 and 8–9, which are well defined in the cryo-EM map and thus determine the register of the transmembrane segments (*Figure 3—figure supplement 1B*). We could further constrain the position of the conserved calcium-binding site, as density between α-helices 6, 7 and 8 coincides with the position of the two bound calcium ions of nhTMEM16 (*Figure 3—figure supplement 1C*). The ion conduction pore is lined by residues located on α-helices 3–7 (*Figure 3*). In contrast to the transmembrane segments close to the dimer interface (i.e. α-helices 1, 2, 9, 10), several of the pore-forming α-helices have changed their position relative to nhTMEM16 (*Figures 1B* and *4*, *Figure 4—figure supplement 1A* and *Videos 3* and *4*). These changes are most pronounced for α-helices 3, 4 and 6. As a consequence of conformational rearrangements, α4 and α6, which line the opposite borders of the membrane-accessible subunit cavity of nhTMEM16, have come into contact at the extracellular part of the membrane to form a protein-enclosed conduit that is shielded from lipids (*Figures 3* and *4*, *Figure 3—figure supplement 1D*, *Figure 4—figure supplement 1A* and *Videos 3* and *4*). Together with α-helices 3,

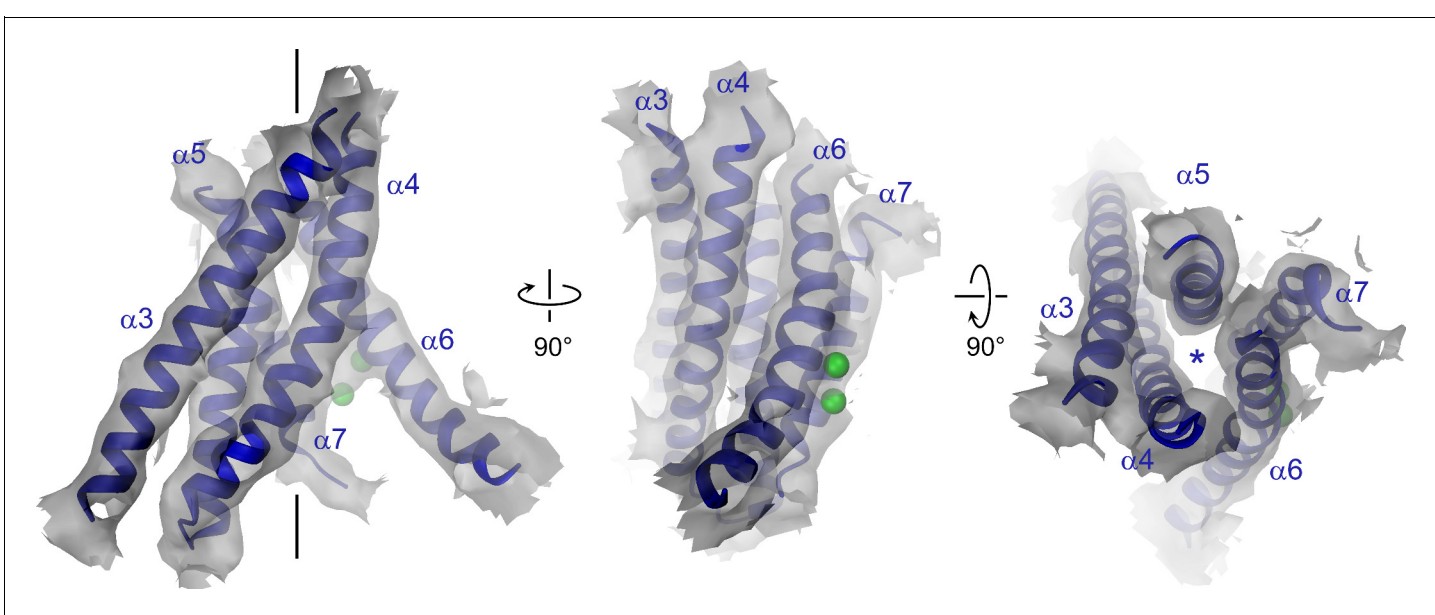

**Figure 3.** Pore region of mTMEM16A. Transmembrane α-helices 3–7 constituting the ion conduction pore of a single mTMEM16A subunit are shown as ribbon and labeled. Sections of the EM density (contoured at 7σ) are superimposed on the model. Green spheres correspond to the positions of bound Ca$^{2+}$ in nhTMEM16. The view in the left panel is as in *Figure 1A*, the relationship of other panels is indicated. The location of the ion conduction pore is marked by a black line (left panel) or an asterisk (right panel).

The following figure supplement is available for figure 3:

**Figure supplement 1.** Pore region and Ca$^{2+}$ binding site.

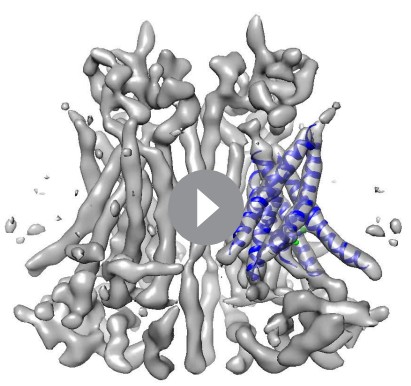

**Video 2.** Pore region of mTMEM16A. Transmembrane α-helices 3–8 constituting the ion conduction pore and the Ca$^{2+}$ binding site of one mTMEM16A subunit (blue). EM density is superimposed. Green spheres correspond to the positions of bound Ca$^{2+}$ in nhTMEM16. The views are as in *Figure 3*.

5 and 7, they constitute the narrow neck of an aqueous pore that spans the extracellular two thirds of the membrane (*Figures 3* and *4*, *Figure 4—figure supplement 1B* and *Video 3*). Towards the intracellular side, the detachment of α4 and α6 results in the a dilation of the pore to a wide intracellular vestibule that is exposed to both the cytoplasm and the lipid bilayer (*Figure 4—figure supplement 1B,C*). The resulting gap between both α-helices may cause a local destabilization of the membrane that is also manifested in a distortion of the detergent micelle observed in the density at lower contour (*Figure 1—figure supplement 4C,D*).

## Functional properties of pore-mutations

A model of the pore is shown in *Figure 5a*. Since the current resolution of the data does not permit a quantitative analysis of its geometry, we restrict our description of the pore to its general geometric features. The wide, intracellular entrance narrows above the region constituting the regulatory Ca$^{2+}$-binding site (*Figure 4—figure supplement 1B*). Under the assumption that the structure is close to a conducting state, the narrow upper part most likely requires permeating ions to shed their hydration shell. This is consistent with the observation that the anion selectivity of TMEM16A follows a type 1 Eisenman sequence (*Qu and Hartzell, 2000*; *Schroeder et al., 2008*; *Yang et al., 2008*), which favors larger anions with a lower solvation energy. The pore is amphiphilic

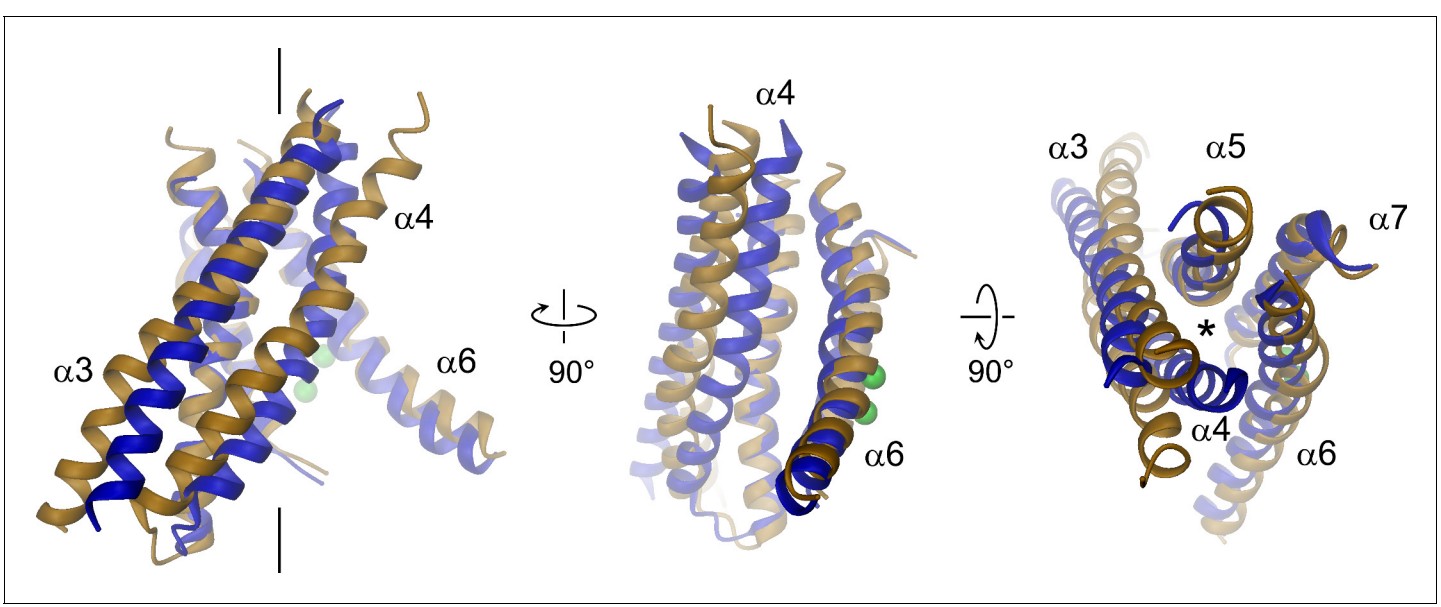

**Figure 4.** Structural relationships between TMEM16 channels and scramblases. Superposition of pore lining helices of mTMEM16A (blue) and nhTMEM16 (beige). Ca$^{2+}$ ions bound to nhTMEM16 are displayed as spheres (green). Views are as in *Figure 3*. The location of the ion conduction pore is marked by a black line (left panel) or an asterisk (right panel).
The following figure supplement is available for figure 4:

**Figure supplement 1.** Pore geometry.

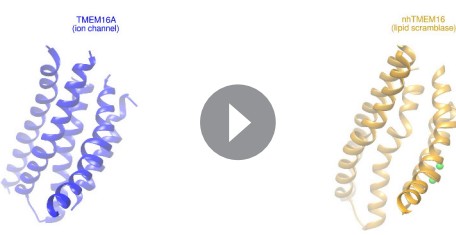

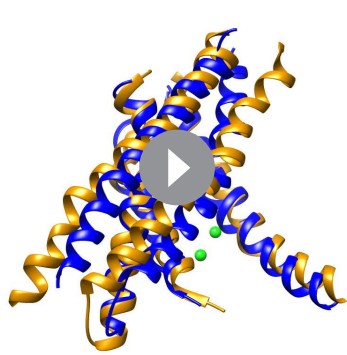

**Video 3.** Comparison of pore regions. Superposition of transmembrane α-helices 3–7 of one subunit of mTMEM16A (blue) and nhTMEM16 (beige). Helices line the ion conduction pore in the channel and the lipid pathway in the scramblase, respectively. Green spheres correspond to the positions of bound Ca²⁺ in nhTMEM16. The views are as in *Figure 4*.

**Video 4.** Helix arrangements in the TMEM16 family. Morph (cyan, middle panel) between transmembrane α-helices 3–7 of one subunit of mTMEM16A (blue, left panel) and nhTMEM16 (beige, right panel). The morph between both structures emphasizes the different arrangement of helices in a lipid scramblase and an ion channel of the TMEM16 family and does not reflect conformational changes in TMEM16A. Helices line the ion conduction pore in the channel and the lipid pathway in the scramblase, respectively. Green spheres correspond to the positions of bound Ca²⁺ in nhTMEM16. The view is similar as in *Figure 4*.

and contains charged, polar and apolar residues. The low effective affinity of Cl⁻ conduction suggests weak interactions with permeating ions (*Figure 5—figure supplement 1A*). Due to the absence of a detailed structural representation of the ion conduction path, we focused on the role of long-range coulombic interactions on anion conduction. We have thus mutated basic residues in the pore to alanine (*Figure 4—figure supplement 1C*) and recorded currents in inside-out patches (*Figure 5—figure supplement 1B*). In these recordings, we can expect deviations from the linear current-voltage relationships of WT in cases where a mutation alters the rate-limiting barriers at either entrance to the narrow part of the pore (*Figure 5—figure supplement 2A*) (*Läuger, 1973*). Such behavior has been previously observed for mutations of Lys 588, where the removal of the positive charge has resulted in a strong outward rectification of the current (*Jeng et al., 2016*; *Lim et al., 2016*). In the model of mTMEM16A, this residue is located at the end of the funnel-shaped vestibule close to the neck of the ion conduction path (*Figure 5A* and *Figure 4—figure supplement 1C*). In our data, the mutation K588A has resulted in a similar rectification, indicating that the truncation of the positively charged side-chain has perturbed the electrostatic interaction with permeating anions, (*Figure 5B* and *Figure 5—figure supplement 2B,C*) effectively increasing the energy barrier of negatively charged ions to enter the pore from its intracellular side (*Figure 5—figure supplement 2A,B*). A similar effect was observed for the nearby mutant K645A, which removes a positive charge from α-helix six at a position that is located slightly further towards the extracellular side (*Figure 5A,C* and *Figure 5—figure supplement 2B,C*). In contrast, several mutations of positively charged residues located in the wide intracellular vestibule did not alter the linear current-voltage relationship of WT (*Figure 5—figure supplement 2C–E*). At the opposite end of the pore, the mutation R535A has resulted in an inward-rectification, indicating that the mutation hampers the entrance of the anion from the outside (*Figure 5A,D* and *Figure 5—figure supplement 2A–C*). In between Lys 645 and Arg 535, the mutation R515A has caused a deviation from the linear current-voltage relationship in both directions (*Figure 5A,E*). Thus, this positive charge most likely lowers a rate-limiting energy barrier for anion permeation halfway through the narrow part of the mTMEM16A pore (*Figure 5—figure supplement 2A,B*). This is consistent with the six-fold lower currents measured for this mutant, despite its robust expression at the surface of HEK cells (*Figure 5—figure supplement 1B, C*). In no case have we seen any change in the reversal potential measured in asymmetric chloride concentrations, which indicates that no single positive charge dominates the strong anion selectivity of the channel (*Figure 5—figure supplement 3*). Together with our structural investigations, the

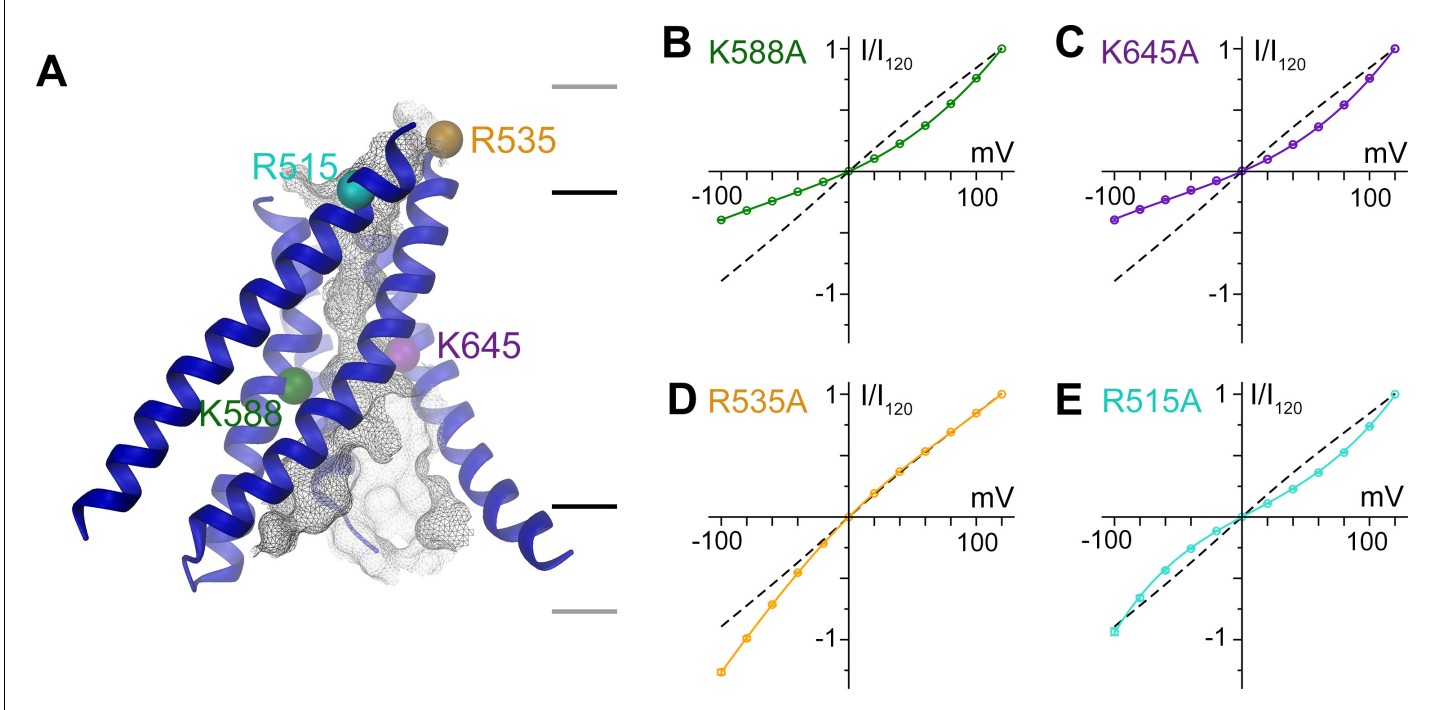

**Figure 5.** Functional properties of mutants of pore lining residues. (**A**) Structure of the pore region of mTMEM16A viewed from within the membrane as shown in the left panel of *Figure 3*. The molecular surface of the ion conduction pore is shown as grey mesh, transmembrane α-helices of the pore region as ribbon, the Cα positions of mutated residues as spheres. Black and grey lines indicate the boundaries of the hydrophobic and polar regions of the bilayer, respectively. I-V relationships of pore mutants (**B**) K588A, (**C**) K645A, (**D**) R535A and (**E**) R515A. Currents were recorded from inside-out patches at 1 mM $Ca^{2+}$ and symmetric $Cl^-$ concentrations. Rundown-corrected data were normalized to the response at 120 mV and show mean and s. e.m. of 8–15 independent recordings. Solid lines show fits to a barrier model. The I-V relationship of WT is shown as dashed line for comparison.

The following figure supplements are available for figure 5:

**Figure supplement 1.** Electrophysiology.

**Figure supplement 2.** Permeation model and properties of pore mutants.

**Figure supplement 3.** Ion selectivity of pore mutants.

electrophysiology data support the notion of a narrow pore in TMEM16A that widens towards the intracellular side.

## Discussion

The present study has addressed structural relationships within the TMEM16 family. Since the majority of TMEM16 proteins work as lipid scramblases, which catalyze the diffusion of lipids between the two leaflets of a bilayer, it was postulated that the few family members functioning as ion channels may have evolved from an ancestral scramblase (*Whitlock and Hartzell, 2016*). However, the way in which TMEM16 channels have adapted to fulfill their distinct functional task has remained unknown. The structure of mTMEM16A reported here has now resolved this question. As anticipated from the strong sequence conservation, the general architecture of each subunit is shared between both branches of the family (*Figure 1B*). A previous structure-based hypothesis suggested a possible subunit rearrangement in dimeric TMEM16 channels, where both subunit cavities come together to form a single enclosed pore (*Brunner et al., 2014*). Although this hypothesis was already refuted by recent functional investigations, which demonstrated that the protein contains two ion conduction pores that are independently activated by $Ca^{2+}$ (*Jeng et al., 2016*; *Lim et al., 2016*), the ultimate

proof for a double barreled channel is now provided by the mTMEM16A structure, which reveals the location of two pores, each contained within a single subunit of the dimeric protein. A different proposition, referred to as the proteolipidic pore hypothesis, postulated that the ion conduction pathway in TMEM16 channels is partly composed of lipids (*Whitlock and Hartzell, 2016*). The authors suggested that immobilized lipid headgroups lining the membrane-exposed ion conduction pore may lower the dielectric barrier for permeating ions on their way across the lipid bilayer (*Whitlock and Hartzell, 2016*). Our study has also provided strong evidence against this hypothesis. Instead, the model of mTMEM16A shows that α-helical rearrangements have resulted in occlusion of the lipid pathway, while opening up a conductive pore which is largely shielded from the membrane (*Figure 6* and *Videos 3* and *4*). The only potential access of lipids is provided on the intracellular side where the detachment of transmembrane α-helices 4 and 6 form a funnel-shaped vestibule that is exposed to the cytoplasm and the lipid bilayer (*Figures 5A* and *6B*). The gap between both α-helices may be a relic of an ancestral scramblase, and as suggested by the observed distortion of the detergent micelle in mTMEM16A, possibly destabilizes the bilayer (*Figure 1—figure supplement 4B,D*). Notably, this gap is also present in nhTMEM16, where a similar effect of membrane-bending has been proposed to facilitate scramblase activity, as suggested by molecular dynamics simulations (*Bethel and Grabe, 2016*). In this respect, it is noteworthy that the intracellular region connecting transmembrane α-helices 4 and 5 has recently been identified to play an important role in lipid scrambling in TMEM16F and was thus assigned the term 'scramblase domain' (*Yu et al., 2015*). Whereas TMEM16A itself does not facilitate lipid transport, scrambling activity was conferred to a chimeric TMEM16A protein carrying the 'scramblase domain' of TMEM16F (*Yu et al., 2015*) or the equivalent region of TMEM16E (*Gyobu et al., 2015*). Although these results emphasize the general role of the intracellular funnel region for lipid interactions, the altered structure of the 'subunit

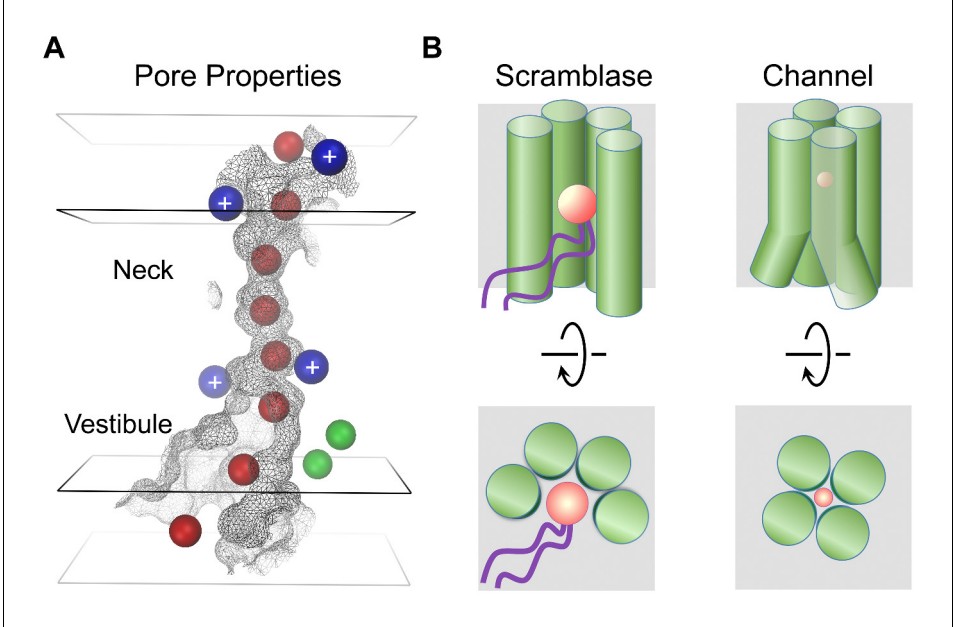

**Figure 6.** Mechanistic relationships within TMEM16 family. (**A**) Depiction of the mTMEM16A pore. The molecular surface of the pore region is shown as grey mesh. The boundaries of hydrophobic (black) and polar regions (grey) of the membrane are indicated by rectangular planes. The positions of positively charged residues affecting ion conduction are depicted as blue and bound $Ca^{2+}$ ions as green spheres. Hypothetical $Cl^-$ ions (radius 1.8 Å) placed along the pore are displayed as red spheres. (**B**) Schematic depiction of features distinguishing lipid scramblases (left) from ion channels (right) in the TMEM16 family. The view is from within the membrane (top panels) and from the outside (bottom panels). The helices constituting the membrane accessible polar cavity in scramblases have changed their location in channels to form a protein-enclosed conduit. A and B, Permeating ions and lipid headgroups are indicated in red.

cavity', in particular the absence of a membrane-exposed polar crevice in TMEM16A, leave the mechanism of lipid scrambling in these chimeras ambiguous.

The structural view of the ion conduction path in mTMEM16A consisting of a funnel-shaped intracellular vestibule that narrows to a tight pore at the extracellular part of the membrane (*Figure 6A*) is supported by our electrophysiology experiments. Analysis of mutants shows minimal influence of basic residues in the wide intracellular vestibule, but pronounced rectification upon similar replacements near the narrow neck of the pore. Remarkably, equivalent mutations of two of these residues (Arg 515 and Lys 645) have previously been described to alter the selectivity between different anions (*Peters et al., 2015*). Assuming that the imaged protein conformation resembles a conducting state, its pore structure suggests that permeating anions have to shed their hydration shell and interact with pore-lining residues (*Figure 6A*). The low effective affinity of Cl$^-$ conduction indicates that there might not be a single strong site for ion coordination, but that the ions might instead weakly interact with the extended pore region (*Qu and Hartzell, 2000*) (*Figure 5—figure supplement 1A*). This is consistent with the fact that no single mutation was identified so far that weakened the strong selectivity for anions over cations (*Figure 5—figure supplement 3*). Although ion conduction was previously also reported for TMEM16 family members which function as lipid scramblases (*Lee et al., 2016*; *Malvezzi et al., 2013*; *Yang et al., 2012*; *Yu et al., 2015*), it was proposed that these processes are leaks accompanying the movement of lipids (*Yu et al., 2015*), which differs significantly from the selective anion permeation described here for TMEM16 channels.

In summary, our work has unraveled how TMEM16 proteins use a similar architecture to exert substantially different functions. Both structures, namely the scramblase nhTMEM16 and the ion channel mTMEM16A, define the structural relationships within the family, whereby a hydrophilic membrane-exposed cavity in TMEM16 scramblases has changed to an aqueous membrane-shielded pore in TMEM16 channels (*Figure 6B* and *Video 4*). Despite the unusual functional breadth of the family, this ligand-gated ion channel turns out to share its mechanism for ion conduction with other, structurally unrelated, channel proteins.

## Material and methods

### Protein expression and purification

A HEK293 cell-line stably expressing the mouse TMEM16A(*ac*) isoform (mTMEM16A, UniProt Q8BHY3.2) containing a 3C cleavage site, a myc- and an SBP-tag at its C-terminus was generated using the Flp-In System (Flp-In-293 Cell Line, R75007, Invitrogen). Adherent HEK cells constitutively expressing mTMEM16A were grown on 10 cm dishes (Corning) at 37°C and 5% CO$_2$ in Dulbecco's modified Eagles's Medium (Sigma) containing either 10% fetal bovine serum (FBS, Sigma) for cell propagation or 5% FBS during protein production. After reaching >80% confluency, cells were harvested by centrifugation at 500 g, washed with PBS buffer (137 mM NaCl, 2.7 mM KCl, 12 mM phosphate pH 7.4) and stored at −20°C until further use. For purification, frozen cell pellets from 7 l of adhesion culture were thawed and resuspended in 140 ml buffer A (20 mM HEPES pH 7.5, 150 mM NaCl and 0.5 mM CaCl$_2$) containing protease inhibitors (cOmplete, Roche). All further steps were carried out at 4°C. Protein was extracted in buffer A containing about 1% digitonin (AppliChem) for 2 hr under gentle agitation. Insoluble material was removed by centrifugation at 22,000 g for 30 min. The supernatant was filtered through a 5 μm filter (Minisart, Sartorius) and incubated with 3 ml of Streptavidin UltraLink resin (Pierce, ThermoScientific) in batch for 1.5 hr. The beads were washed with 60 column volumes of buffer A containing 0.12% digitonin (Calbiochem; buffer B) and eluted with three column volumes of buffer B containing 4 mM of biotin. Protein was deglycosylated for 2 hr by addition of PNGaseF, and subsequently concentrated (Amicon Ultra, 100 k). The concentrated sample was applied to a Superdex 200 size-exclusion chromatography column equilibrated in buffer B. The following day fractions containing target protein were concentrated to obtain 15 μl of pure protein at a final concentration of 3 mg ml$^{-1}$ and subsequently used for EM sample preparation.

### Electron microscopy sample preparation and imaging

2.5 μl of purified mTMEM16A at a concentration of 3 mg ml$^{-1}$ were pipetted onto glow-discharged 200 mesh gold Quantifoil R1.2/1.3 holey carbon grids (Quantifoil). Grids were blotted for 2–5 s with a blotting force of 1 at 20°C and 100% humidity, and flash-frozen in liquid-ethane using an FEI

Vitrobot Mark IV (FEI). Cryo-EM data were collected on a 300 kV FEI Titan Krios electron microscope using a post-column quantum energy filter (Gatan) with a 20 eV slit and a 100 μm objective aperture. Data were collected in an automated fashion on a K2 Summit detector (Gatan) set to super-resolution mode with a pixel size of 0.675 Å and a defocus range of −0.5 to −3.8 μm using SerialEM (*Mastronarde, 2005*). Images were recorded for 15 s with an initial sub-frame exposure time of 300 ms (50 frames total) with a dose of 1.5 e⁻/Å²/frame, and later with a sub-frame exposure time of 150 ms (100 frames total) with a dose of 0.8 e⁻/Å²/frame, resulting in a total accumulated dose on the specimen level of approximately 80 e⁻/Å².

## Image processing

A total of 5503 dose-fractionated super-resolution images were 2 × 2 down-sampled by Fourier cropping (final pixel size 1.35 Å) and subjected to motion correction and dose-weighting of frames by MotionCor2 (*Zheng et al., 2016*). The contrast transfer function (CTF) parameters were estimated on the movie frames by ctffind4.1 (*Rohou and Grigorieff, 2015*). Images showing a strong drift, higher defocus than −3.8 μm or a bad CTF estimation were discarded, resulting in 4178 images used for further analysis. Image processing was performed using the software package RELION1.4 (*Scheres, 2012*) and at a later stage RELION2.0 (*Kimanius et al., 2016*). Approximately 4000 particles were manually picked to generate templates for automated particle selection. Following automated picking in RELION, false positives were eliminated manually or through a first round of 2D classification resulting in 755,348 particles. These were subjected to several rounds of 2D classification to remove particles belonging to low-abundance classes. The remaining 522,701 particles were sorted during 3D Classification with C2 symmetry imposed. A model was generated from the nhTMEM16 X-ray structure (*Brunner et al., 2014*) (PDBID 4WIS), low-pass filtered to 60 Å and used for the first round of classification. In an iterative mode, the best output map was used for subsequent classification or refinement rounds. Similar classes, comprising 377,371 particles, were combined and subjected to auto-refinement in RELION. The resulting map was masked and had a resolution of 7.35 Å. To further improve the quality of the density map, per-particle alignment of the frames was performed using the polishing algorithms in RELION. The best results were obtained upon inclusion of all dose-weighted frames and application of a running average window of 9, a standard deviation of 2 pixels on the translations during movie refinement and 200 pixels on particle distance during particle polishing (*Scheres, 2014*). Polished particles were subjected to another round of 2D and 3D classification, resulting in a selection of 213,243 particles. The final polished, auto-refined and masked map had a resolution of 6.6 Å. The final map was sharpened using an isotropic b-factor ranging between −351 Å² and −700 Å² and used for model building. Local resolution estimates were calculated within RELION. All resolutions were estimated using the 0.143 cut-off criterion (*Rosenthal and Henderson, 2003*) with gold-standard Fourier shell correlation (FSC) between two independently refined half maps (*Scheres and Chen, 2012*) (*Figure 1—figure supplement 3B*). During post-processing, the approach of high-resolution noise substitution was used to correct for convolution effects of real-space masking on the FSC curve (*Chen et al., 2013*).

## Model building

A poly-alanine model encompassing the secondary structure elements of mTMEM16A was constructed based on the nhTMEM16 X-ray structure (*Brunner et al., 2014*) (PDBID 4WIS). For that purpose the nhTMEM16 structure was initially docked into the EM density using UCSF Chimera (*Pettersen et al., 2004*). The fit of certain fragments as rigid bodies was subsequently improved in Coot (*Emsley and Cowtan, 2004*). Long and poorly conserved loop regions and side-chains were removed from the model and residues of mTMEM16A were assigned based on a sequence alignment (*Figure 1—figure supplement 1*). Density for conserved short loops and bound Ca²⁺ ions assisted the assignment of the register for residues of the pore region. The structure was improved locally by real space refinement in Coot (*Emsley and Cowtan, 2004*) followed by global real space refinement in Phenix (*Adams et al., 2002*; *Afonine et al., 2013*) maintaining strong secondary structure and symmetry constraints between the two subunits of the dimeric protein (*Table 1*). The final model consists of 434 residues and includes the β-strands and α-helices of the N-terminal domain, two peripheral and 10 transmembrane spanning α-helices of the TMD, including short and conserved loop regions, and the first α-helix of the C-terminal domain. It contains residues 123–127,

167–214, 242–254, 278–282, 295–305, 315–355, 409–438, 486–520, 535–602, 633–666, 681–781, 855–885 and 892–904. The molecular surface of the pore was calculated with MSMS (*Sanner et al., 1996*) from coordinates where side-chain positions of residues constituting the ion conduction pore were modeled in Coot (*Emsley and Cowtan, 2004*). Model building was performed on the final cryo-EM map sharpened with a B-factor of −700 Å$^2$, as shown in all figures except for *Figure 1—figure supplement 4C,D* where a B-factor of −351 Å$^2$ was applied. All structure calculations and model building were performed using software compiled by SBGrid (*Morin et al., 2013*). Structure figures ad movies were prepared with DINO (http://www/dino3d.org) or UCSF Chimera (*Pettersen et al., 2004*).

## Electrophysiology

For electrophysiology, the mTMEM16A(*ac*) cDNA was cloned into a pcDNA3.1 plasmid modified for the FX-system (*Geertsma and Dutzler, 2011*) with a C-terminal YFP/SBP/myc tag. Mutations were introduced by a modified QuikChange method (*Zheng et al., 2004*) and confirmed by sequencing. HEK293T cells (ATCC CRL-1573) were transfected with 3 µg of DNA per 6 cm dish using the calcium phosphate precipitation method. Transfected cells were used within 24 to 96 hr after transfection. Inside-out patches were excised from HEK293T cells expressing WT or mutant constructs after the formation of a gigaohm seal. The seal resistance was typically 4–8 GΩ or higher. Recording pipettes were pulled from borosilicate glass capillaries (O.D. 1.5 mm, I.D. 0.86 mm (Sutter)) and fire-polished with a microforge (Narishige) before use. Pipette resistance was 3–8 MΩ when filled with recording solution. Voltage-clamp recordings were performed using the Axopatch 200B amplifier controlled by the Clampex 10.6 software through Digidata 1550 (Molecular Devices). Raw signals were analogue-filtered at 5 kHz through the in-built 4-pole Bessel filter and digitized at 20 kHz. Liquid junction potential was not corrected. Solution exchange was performed using a theta glass pipette mounted on a high-speed piezo switcher (Siskiyou).

Experiments were performed at 1 mM Ca$^{2+}$ on the intracellular side to maximize channel activation. This also minimizes interference by time-dependent relaxation of the current during a voltage step when information on the instantaneous current response is required. The pipette solution contained 150 mM NaCl, 5.99 mM Ca(OH)$_2$, 5 mM EGTA and 10 mM HEPES at pH 7.4 (NaCl buffer). Rectification experiments were carried out under symmetrical ionic conditions with a bath solution having the same composition as the pipette solution. For permeability experiments, the NaCl concentration was adjusted by mixing the NaCl buffer with a (NMDG)$_2$SO$_4$ solution containing 100 mM (NMDG)$_2$SO$_4$, 5.99 mM Ca(OH)$_2$, 5 mM EGTA and 10 mM HEPES at pH 7.4 at the required ratio. For high ionic strength, KCl buffer, containing 150 mM KCl, 5.99 mM Ca(OH)$_2$, 5 mM EGTA and 10 mM HEPES at pH 7.4, was used for both bath and pipette solutions to minimize the junction potential. For concentrations above 150 mM Cl$^-$, KCl was dissolved in this solution at the required amounts.

Data were background-subtracted before analysis. Background current was obtained by recording in the corresponding solution in the absence of intracellular Ca$^{2+}$. I-V data were obtained by measuring the instantaneous current after each voltage jump in a step protocol (*Figure 5—figure supplement 1B*). To correct for current rundown, the measured instantaneous currents were divided by the fraction of current remaining during the pre-pulse at +80 mV and were expressed as normalized current (I/I$_{120mV}$). This is important as uncorrected current rundown can give rise to artificial rectification. Potential voltage offset was detected by recording in symmetrical solutions. Only patches with an offset <2 mV were accepted for analysis. The voltage offset was subtracted from the reversal potentials obtained from asymmetric ionic conditions for the same patch whenever possible. This was not possible for a minority of constructs that displayed low current and/or fast rundown. For these constructs, the averaged offset was subtracted from the averaged reversal potentials obtained in asymmetric ionic conditions. Data are presented as mean ± s.e.m..

## Model of permeation

To analyze the position-dependent effect of mutations on the rectification of the current, we have employed a barrier model akin to that described by *Läuger (1973)*. We are aware of the general limitations of barrier models for quantitative interpretations (*Eisenberg, 1999*) and thus only aim for a phenomenological description. The model assumes the presence of multiple hypothetical energy

barriers on the ion conduction path that are not necessarily identical (Appendix *Scheme 1*). The equation used to fit the experimental I-V data and to determine the descriptive energy profile of the constructs is shown below.

$$I = zFAe^{\frac{zFV}{2nRT}} \frac{c_i - c_o e^{-\frac{zFV}{RT}}}{e^{-zFV\frac{n-1}{nRT}} + \left(\frac{1}{\sigma_h}\right)\frac{1-e^{-zFV\frac{n-2}{nRT}}}{e^{\frac{zFV}{nRT}}-1} + \frac{1}{\sigma_\beta}}$$

The model contains three free parameters (n, $\sigma_\beta$ and $\sigma_h$) that govern the shape of the I-V relation, which, with reasonable constraints, can be reliably determined from our data (*Figure 5B–E* and *Figure 5—figure supplement 2A,B*). A is a proportionality factor, n is the number of barriers and $\sigma_\beta$ and $\sigma_h$ are relative rates for outward flux across the innermost and internal barriers compared to the external barrier. For our fit, we used three barriers to describe the observed behavior and determined $\sigma_\beta$ and $\sigma_h$ for the mutant constructs. The relative increase of the barrier height is obtained by

$$\Delta E_{a\,(in-out)} = -RTln\sigma_\beta$$

$$\Delta E_{a\,(mid-out)} = -RTln\sigma_h$$

where $E_a$ is the activation energy corresponding to the respective rate constant. These parameters were used to construct descriptive energy profiles to illustrate the effect of the mutations and are shown in *Figure 5—figure supplement 2A,B*. For more details, see Appendix 1.

## Accession codes

The electron density map has been deposited in the Electron Microscopy Data Bank under the accession code EMD-3658 and the coordinates of the model in the Protein Data Bank under the accession code 5NL2.

## Acknowledgements

We thank O Medalia and M Eibauer, the center for microscopy and image analysis (ZMB) of the University of Zurich, and the Mäxi foundation for the support and access to the Electron Microscopes. Novandy K Lim is acknowledged for advice in purification, Justin D Walter for comments on the manuscript and all members of the Dutzler lab for help at various stages of the project.

## Additional information

### Funding

| Funder | Grant reference number | Author |
|---|---|---|
| Seventh Framework Programme | 339116 AnoBest | Raimund Dutzler |
| Universität Zürich | Forschungskredit FK-16-036 | Cristina Paulino |

The funders had no role in study design, data collection and interpretation, or the decision to submit the work for publication.

### Author contributions

CP, Conceptualization, Funding acquisition, Data curation, Investigation, Validation, Methodology, Writing—original draft, Writing—review and editing, Planned experiments, prepared the sample for cryo-EM, collected EM data and proceeded with structure determination; YN, Methodology, Writing—review and editing, Expressed and purified protein; AKML, Conceptualization, Data curation, Investigation, Validation, Methodology, Writing—original draft, Writing—review and editing, Generated mutants, performed electrophysiological recordings and fitted data; VK, Methodology, Writing—review and editing, Expressed and purified protein ; JDB, SS, Resources, Generated the stable HEK293 cell-line expressing TMEM16A; RD, Conceptualization, Supervision, Funding acquisition,

Validation, Investigation, Visualization, Writing—original draft, Writing—review and editing, Project administration

## Author ORCIDs

Cristina Paulino, http://orcid.org/0000-0001-7017-109X
Andy KM Lam, http://orcid.org/0000-0002-2983-3044
Valeria Kalienkova, http://orcid.org/0000-0002-4143-6172
Raimund Dutzler, http://orcid.org/0000-0002-2193-6129

## Additional files

### Major datasets

The following datasets were generated:

| Author(s) | Year | Dataset title | Dataset URL | Database, license, and accessibility information |
|---|---|---|---|---|
| Cristina Paulino, Yvonne Neldner, Andy KM Lam, Valeria Kalienkova, Janine Denise Brunner, Stephan Schenck, Raimund Dutzler | 2017 | cryo-EM structure of the mTMEM16A ion channel at 6.6 A resolution | https://www.ebi.ac.uk/pdbe/entry/emdb/EMD-3658 | Publicaly available at the EMBL-EBI Protein Data Bank (accession no: EMD-3658) |
| Cristina Paulino, Yvonne Neldner, Andy KM Lam, Valeria Kalienkova, Janine Denise Brunner, Stephan Schenck, Raimund Dutzler | 2017 | cryo-EM structure of the mTMEM16A ion channel at 6.6 A resolution | http://www.rcsb.org/pdb/explore/explore.do?structureId=5NL2 | Publicly available at the RCSBProtein Data Bank (accession no: 5NL2) |

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

## Appendix 1

### A model to describe the position-dependent effect of mutations in the mTMEM16A pore

To analyze the position-dependent effect of mutations on the rectification of the current, we have employed a barrier model akin to that described by Läuger (*Läuger, 1973*). We assumed that there is no saturation of the pore as the Cl⁻ concentration we used was more than two times lower than the apparent $K_M$ value (*Figure 5—figure supplement 1A*). Because the apparent $K_M$ value is high (>300 mM), it is also safe to assume that no deep wells exist along the pore. We thus assumed that all wells in the model have the same energy level as the external and internal media, which also implies that at zero voltage the forward and backward rate constants for the same barrier are identical. We further assumed that the barriers are evenly distributed along the channel so that each of the rate constants have the same voltage dependence with the corresponding polarity (*Scheme 1*).

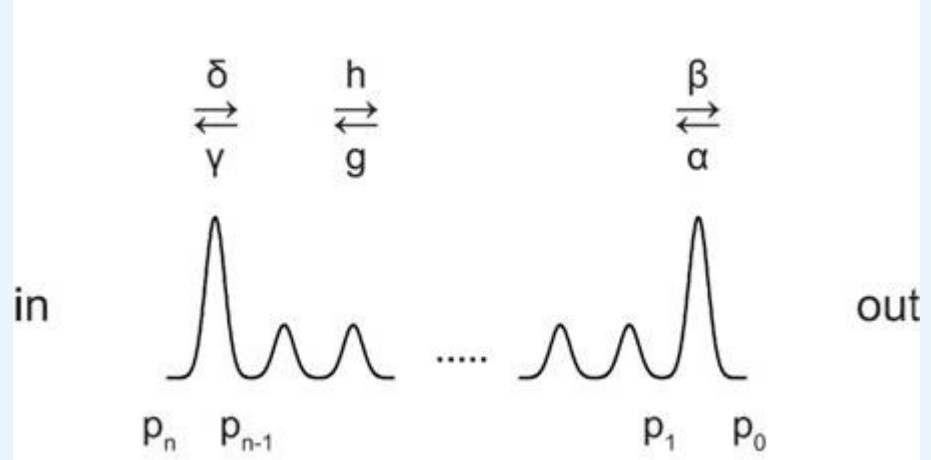

**Scheme 1.** Depiction of the free energy profile of ions permeating the pore of an ion channel.

In this model, $\beta$ is the rate constant of the outermost barrier and the rate constant of the innermost barrier is defined as $\delta = \sigma_\beta\beta$. The rate constants of the barriers in between are assumed to be identical and are defined as $h = \sigma_h\beta$. These rate constants $\beta$, $\delta = \sigma_\beta\beta$ and $h = \sigma_h\beta$ describe the rate of flux from the intracellular to the extracellular side and the corresponding reverse rate constants are $\alpha$, $\gamma$ and g. $c_o$ and $c_i$ are the ion concentrations on the extracellular and the intracellular sides respectively. $v$ is a proportionality factor that has a dimension of volume and may be interpreted as the hypothetical volume for outermost well at the channel entrance. The number of barriers is n. $p_i$'s are the probabilities of occupying the $i^{th}$ wells (from $p_0$ to $p_n$). The voltage dependence of the rate constants from the inside to the outside is $e^{\frac{zFV}{2nRT}}$, where V is the membrane potential, z is the valence of the ion, and R, T and F have their usual meanings. The flux (J) equations are

$$J = -\alpha v c_o + \beta p_1 \quad , \quad p_1 = \phi v c_o + \frac{J}{\beta}$$

$$J = -g p_1 + h p_2 \quad , \quad p_2 = \phi p_1 + \frac{J}{h} = \phi\left(\phi v c_o + \frac{J}{\beta}\right) + \frac{J}{h}$$
$$= \phi^2 v c_o + \phi\frac{J}{\beta} + \frac{J}{h}$$

where $\phi = \frac{\alpha}{\beta} = \frac{g}{h} = \frac{\gamma}{\delta} = e^{-\frac{zFV}{nRT}}$

$$J = -gp_2 + hp_3 \quad , \quad p_3 = \phi p_2 + \frac{J}{h} = \phi\left(\phi^2 vc_o + \phi\frac{J}{\beta} + \frac{J}{h}\right) + \frac{J}{h}$$
$$= \phi^3 vc_o + \phi^2\frac{J}{\beta} + \frac{J}{h}(\phi + 1)$$

$$J = -gp_3 + hp_4 \quad , \quad p_4 = \phi p_3 + \frac{J}{h} = \phi\left(\phi^3 vc_o + \phi^2\frac{J}{\beta} + \frac{J}{h}(\phi + 1)\right) + \frac{J}{h}$$
$$= \phi^4 vc_o + \phi^3\frac{J}{\beta} + \frac{J}{h}(\phi^2 + \phi + 1)$$

In general,

$$p_i = \phi^i vc_o + \phi^{i-1}\frac{J}{\beta} + \frac{J}{h}\sum_0^{i-2}\phi^j = \phi^i vc_o + \phi^{i-1}\frac{J}{\beta} + \frac{J}{h}\left(\frac{1-\phi^{i-1}}{1-\phi}\right)$$

For the (n-1)$^{\text{th}}$ and n$^{\text{th}}$ wells,

$$p_{n-1} = \phi^{n-1} vc_o + \phi^{n-2}\frac{J}{\beta} + \frac{J}{h}\left(\frac{1-\phi^{n-2}}{1-\phi}\right)$$

$$vc_i = p_n = \phi p_{n-1} + \frac{J}{\delta} = \phi\left(\phi^{n-1} vc_o + \phi^{n-2}\frac{J}{\beta} + \frac{J}{h}\left(\frac{1-\phi^{n-2}}{1-\phi}\right)\right) + \frac{J}{\delta}$$

$$vc_i = \phi^n vc_o + \phi^{n-1}\frac{J}{\beta} + \frac{J}{h}\phi\left(\frac{1-\phi^{n-2}}{1-\phi}\right) + \frac{J}{\delta}$$

$$J = \frac{v(c_i - \phi^n c_o)}{\phi^{n-1}\frac{1}{\beta} + \frac{1}{h}\phi\left(\frac{1-\phi^{n-2}}{1-\phi}\right) + \frac{1}{\delta}}$$

Expressing h and δ as $\sigma_h\beta$ and $\sigma_\beta\beta$ respectively,

$$J = \frac{v(c_i - \phi^n c_o)}{\phi^{n-1}\frac{1}{\beta} + \frac{1}{\sigma_h\beta}\phi\left(\frac{1-\phi^{n-2}}{1-\phi}\right) + \frac{1}{\sigma_\beta\beta}}$$

$$J = \frac{\beta v(c_i - \phi^n c_o)}{\phi^{n-1} + \frac{1}{\sigma_h}\phi\left(\frac{1-\phi^{n-2}}{1-\phi}\right) + \frac{1}{\sigma_\beta}}$$

Substituting back the exponential terms and converting into current (I) by multiplying by zF,

$$I = zFAe^{\frac{zFV}{2nRT}}\frac{c_i - c_o e^{-\frac{zFV}{RT}}}{e^{-zFV\frac{n-1}{nRT}} + \left(\frac{1}{\sigma_h}\right)\frac{1-e^{-zFV\frac{n-2}{nRT}}}{e^{\frac{zFV}{nRT}}-1} + \frac{1}{\sigma_\beta}}$$

This equation was used to fit the experimental I-V data and to determine the descriptive energy profile of the constructs. The model contains three free parameters (n, $\sigma_\beta$ and $\sigma_h$) that govern the shape of the I-V relation, which, with reasonable constraints (see below), can be reliably determined from our data (*Figure 5B–E* and *Figure 5—figure supplement 2A, B*). $A = \beta_0 v$ is a proportionality factor where $\beta_0$ is the value of $\beta$ at V = 0. Of note, when

n = 2, the $\left(\frac{1}{\sigma_h}\right)\frac{1-e^{-zFV\frac{n-2}{nRT}}}{e^{\frac{zFV}{nRT}}-1}$ term vanishes, and the equation is reduced to the case of only two barriers. It can also be seen that this equation can be reduced to the Nernst equation at zero current.

The general features of the model are shown in **Appendix 1—figure 1**. In the simplest case where the energy profile consists of only the δ and β barriers without the h barriers, increasing n causes the non-linearly increasing conductance of the current at both ends of the I-V curves to saturate while the conductance at zero voltage remains unchanged (**Appendix 1—figure 1A, B**). The introduction of h barrier(s) with height(s) identical to the δ and β barriers results in a decrease in the overall current amplitude and a non-linear increase in the conductance with increasing voltage (**Appendix 1—figure 1C, D**). This non-linear increase in the conductance eventually flattens as n approaches infinity and the resulting I-V curves become ohmic (**Appendix 1—figure 1D**, inset). This indicates that, for any value of n, the presence of h barrier(s) of significant height relative to the δ and β barriers cannot result in bell-shaped conductance-V curves. As $\sigma_h$ increases, i.e. the height of the h barriers decreases, however, the non-linearly increasing conductance-V curves revert to bell-shaped curves (**Appendix 1—figure 1E, F**).

We tested this model first by fitting the I-V curve of WT at symmetrical 150 mM Cl⁻. The conductance-V relation obtained from the interpolated I-V curve is bell-shaped (**Appendix 1—figure 2A, B**), which is consistent with n > 2 and small h barrier(s) relative to the δ and β barriers (**Appendix 1—figure 1**). The imperfect symmetry indicates asymmetry in the barrier heights on the ends of the energy profile. However, when all parameters were allowed to vary the fitting did not converge and the resulting parameter estimates had very wide 95% confidence intervals. Nonetheless, the qualitative agreement with the calculated model behavior allowed us to constrain $\sigma_h$ to reasonable values and/or to omit the h barrier (s) completely. Such measure allowed us to obtain a reasonable value for n, which was estimated to be 2.8 (**Appendix 1—figure 2A**). Again, assuming large $\sigma_h$, we fitted the family of I-V curves of WT below 150 mM Cl⁻ globally and obtained a reasonable agreement with the model and an estimate of 3.1 for n (**Appendix 1—figure 2C**). The value of n was therefore chosen to be three and was subsequently used as a fixed parameter to determine the relative rates $\sigma_\beta$ and $\sigma_h$ for the mutant constructs. The relative increase of the barrier height is obtained by

$$\Delta E_{a\,(in-out)} = -RTln\sigma_\beta$$

$$\Delta E_{a\,(mid-out)} = -RTln\sigma_h$$

where $E_a$ is the activation energy corresponding to the respective rate constant. These parameters were used to construct descriptive energy profiles to illustrate the effect of the mutations and are shown in **Figure 5—figure supplement 2A,B**. We have arbitrarily placed the energy profiles relative to the innermost or the outermost barrier for illustrative purpose. This most likely has little effect on the energy profiles of the mutants where the mutated residue is located on the peripheral ends (R535A, K588A and K645A) as these mutations are unlikely to affect the energy barrier on the opposite end of the pore. This may not be the case, however, for the R515A mutant as this residue lies in the middle of the pore. The small current observed for this mutant (**Figure 5—figure supplement 1B**) implies that the whole energy profile might in fact be shifted upwards compared to that of the WT. This might be expected if the effect of R515 is of coulombic nature, and like the other mutations in the pore mutation of which may elevate nearby energy barriers especially in a narrow pore. Nonetheless, the general observation is that as the mutation moves along the pore, its effect on local ion conduction/barrier height shifts accordingly (**Figure 5—figure supplement 2B**), which is reflected on their macroscopic conduction properties as observed in our experiments (**Figure 5B–E**).

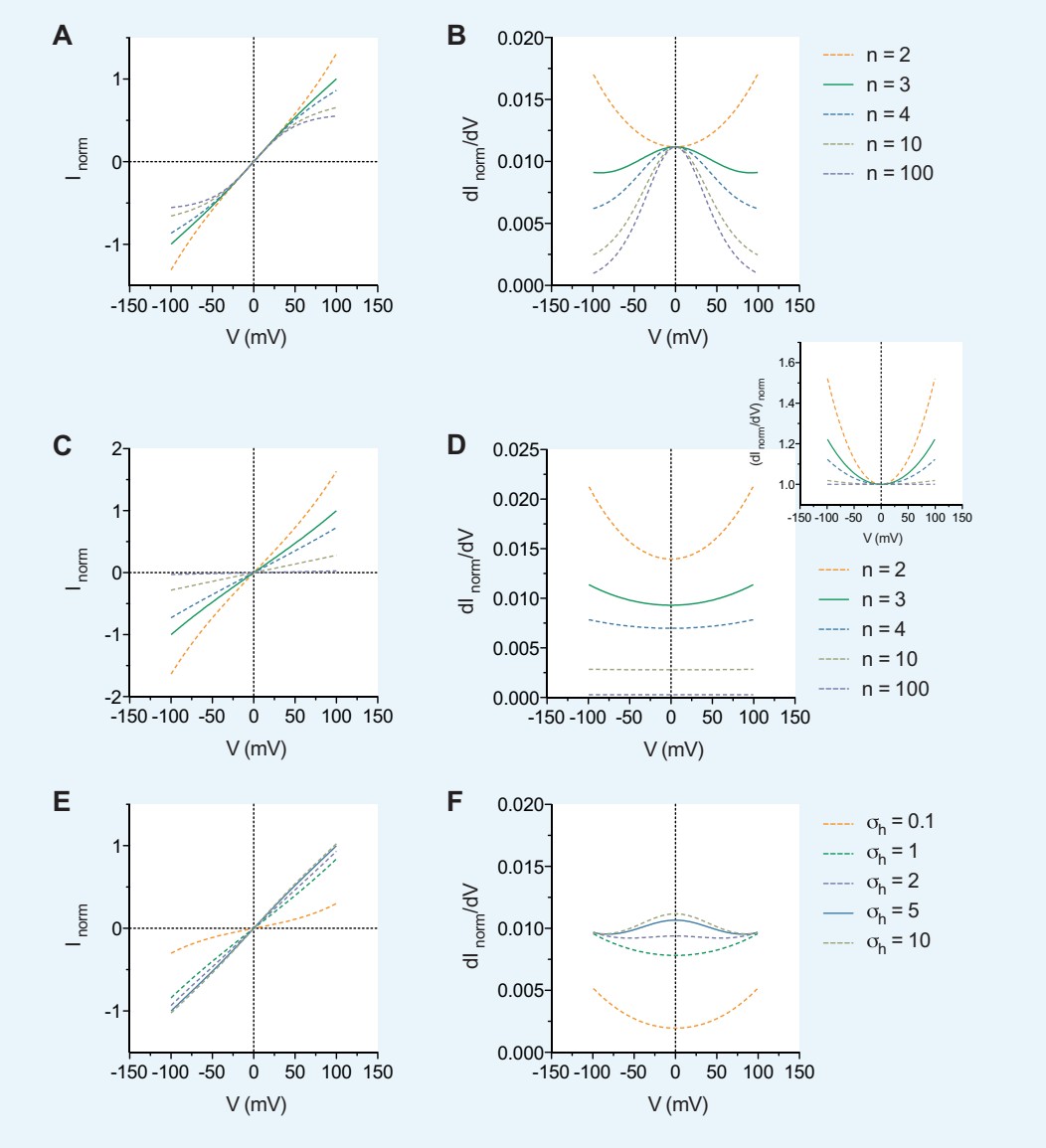

**Appendix 1—figure 1.** General features of the model. (**A**) I-V and (**B**) conductance-V curves calculated with $\sigma_\beta = 1$ with the $\left(\frac{1}{\sigma_h}\right)\frac{1-e^{-zFV\frac{n-2}{nRT}}}{e^{\frac{zFV}{nRT}}-1}$ term omitted and the indicated values of n. (**C**) I-V and (**D**) conductance-V curves calculated with $\sigma_\beta = \sigma_h = 1$ and the indicated values of n. Inset, as in D but normalized to the minima of the curves. (**E**) I-V and (**F**) conductance-V curves calculated with n = 3, $\sigma_\beta = 1$ and the indicated values of $\sigma_h$. The curves are relative to the curve where n = 3 (**A**, **B**, **C** and **D**) and $\sigma_h = 5$ (**E** and **F**).

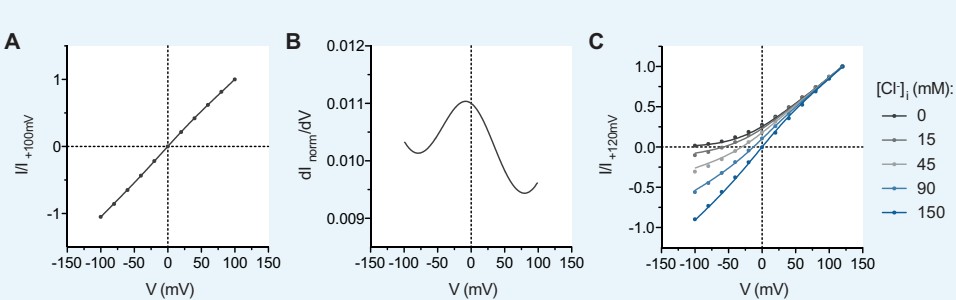

**Appendix 1—figure 2.** Parameter estimation. (**A**) I-V relation of WT mTMEM16A at symmetrical 150 mM Cl⁻. The curve is a fit to the model with the $\left(\frac{1}{\sigma_{\mathrm{h}}}\right)\frac{1-e^{-zFV\frac{n=2}{nRT}}}{e^{\frac{zFV}{nRT}}-1}$ term omitted. n = 2.79 (2.60 to 2.98), $\sigma_{\beta}$ = 1.06 (1.03 to 1.09). (**B**) Conductance-V relation of WT mTMEM16A calculated from the interpolated I-V curve shown in A. (**C**) Family of I-V curves of WT mTMEM16A at 150 mM extracellular Cl⁻ and the indicated intracellular concentrations of Cl⁻. The curves are a global fit to the model with the $\left(\frac{1}{\sigma_{\mathrm{h}}}\right)\frac{1-e^{-zFV\frac{n=2}{nRT}}}{e^{\frac{zFV}{nRT}}-1}$ term omitted. n = 3.14 (3.09 to 3.20), $\sigma_{\beta}$ = 1.08 (1.07 to 1.09). Data in this figure are as in *Figure 5—figure supplement 3*. The values in parentheses indicate the 95% confidence interval of the parameter estimate.

