## [Decision Letter]

Thank you for submitting your article "Structural basis for anion conduction in the calcium-activated chloride channel TMEM16A" for consideration by *eLife*. Your article has been reviewed by three peer reviewers, and the evaluation has been overseen by a Reviewing Editor and Richard Aldrich as the Senior Editor. The following individuals involved in review of your submission have agreed to reveal their identity: H Criss Hartzell (Reviewer #1); Youxing Jiang (Reviewer #2); Sjors HW Scheres (Reviewer #3).

The reviewers have discussed the reviews with one another and the Reviewing Editor has drafted this decision to help you prepare a revised submission.

Summary:

This manuscript describes the structure of the mammalian Ca-activated Cl channel TMEM16A determined by cryo-EM. The authors conclude that the structure differs in a significant way from another member of the TMEM16 superfamily that is a phospholipid scramblase. First, the mechanism of dimerization appears to be modified compared to the nhTMEM16 scramblase. Second and most notably, the furrow formed mainly by helices 4-6 has be reconfigured so that helix 4 now encloses the conduction pathway to form a protein-lined channel rather than a groove. If this is true, this is a very significant finding. The TMEM16 family has been the subject of intense interest and scrutiny recently and this paper adds significant new information. The work is carefully performed and beautifully presented. Nevertheless, reviewers have raised some concerns most of which we believe can be addressed by carefully underscoring the limitations of your data.

Essential revisions:

1) I have been struggling to understand whether the protein is in a conducting or non-conducting state. The authors assume a conducting state because it was purified in the presence of Ca. However, it is well-known that TMEM16A inactivates in the presence of high Ca. This has been shown by the Dutzler lab as well as others. In our hands, TMEM16A enters a non-conducting state within several minutes in the presence of 0.5 mM Ca – the concentration used here for purification. Furthermore, it is not clear to me that the "pore" of the cryo structure is large enough to accommodate a Cl ion, not to mention larger ions like iodide (radius 2.2Å) that is more permeant than Cl. It seems that there is barely enough space for a Cl ion to move through the pore of the Ala model. At the level of residues 546, 595, and 641 there is a constriction that is barely large enough to accommodate a Cl ion. With the significantly larger sidechains of the real amino acids at this location: N546, P595, and I641, there does not seem to be enough space for a Cl ion to pass. To get some insight into this, I replaced the Ala residues with the TMEM16A amino acids using the most common rotamer for each position and then energy minimized the structure. I understand that this structure could be completely bogus, but the aperture is >1Å too small in places to accommodate a Cl ion. This suggests that either this is a non-conducting state or that the conduction pathway is not located in the cavity between helices 4-6. I appreciate that the cryo structure may not have the resolution to allow the conclusions that I am suggesting, but yet the possibility that this is a non-conducting structure seems a possibility that must be addressed.

2) The mutagenesis experiments tend to support the idea that the Cl conduction pathway is the in the furrow (now a pore), but the residues chosen for mutation are located at the extreme ends of the enclosed pore (the α carbons of R515 and K588 are ~20 Å apart). While most of the residues that intervene between the vestibules are hydrophobic and perhaps not ideal candidates for mutagenesis analysis, it would be nice to have data on some amino acids that are located in the middle of the presumptive conduction pathway. The effect of the R515A mutation in particular is relatively small. Given the timeframe for revised submission, if additional experiments are not possible, the authors should discuss the limitations of their existing data.

3) The obtained resolution of 6.6Å from 4,000 K2-recorded micrographs is somewhat disappointing, and raises the question why no higher resolution was obtained. The 2D class averages look somewhat suboptimal. Still, the detailed data acquisition and image processing procedures described all appear to be performed to a high standard. Perhaps the protein sample itself wasn't optimal? Also, perhaps the sample micrograph in Figure 1—figure supplement 1 looks very/too contrasted. Could it be that the area imaged was too thin and the sample had dried too much? Anyway, these are not questions for the authors to answer now, but suggestion to consider for a next structure.

It is refreshing to see that the authors are well aware of its limitations, and are adequately careful in its interpretation. From the cryo-EM point-of-view, this paper is thereby sound and there are no technical reasons to reject it.

4) Discussion section: "[…] as suggested by the observed distortion of the detergent micelle in mTMEM16A". The distortion is not at all clear from the figure. A better figure is needed to support this part of the discussion.

5) Subsection “Image processing”: "Overfitting was further prevented by correcting the final FSC curve for the effect of a soft mask using high-resolution noise substitution" This is not correct. The high-resolution noise substitution at this point is not to prevent overfitting, but rather to correct for convolution effects of real-space masking on the FSC curve.

---

## [Author Response]

*Essential revisions:*

*1) I have been struggling to understand whether the protein is in a conducting or non-conducting state. The authors assume a conducting state because it was purified in the presence of Ca. However, it is well-known that TMEM16A inactivates in the presence of high Ca. This has been shown by the Dutzler lab as well as others. In our hands, TMEM16A enters a non-conducting state within several minutes in the presence of 0.5 mM Ca – the concentration used here for purification.*

The reviewer refers to the irreversible rundown of currents observed in patch-clamp recordings of TMEM16A at high Ca^2+^ concentrations. The mechanism for this process is currently unclear and it has also not been shown whether a similar process would happen for the purified protein in detergent solution. Although we think that the structure described in our manuscript shows the general features of a conformation that is close to a conducting state we cannot state this with certainty and have thus introduced the following changes in the manuscript.

Subsection “mTMEM16A structure”:

“Due to the presence of Ca^2+^, it likely shows the channel in a Ca^2+^-bound conformation. In light of the irreversible rundown of TMEM16A-mediated currents observed in patch-clamp experiments at high Ca^2+^ concentrations it is at this point ambiguous whether this conformation corresponds to a conducting or a non-conducting state of the channel.”

Discussion section:

“Assuming that the imaged protein conformation resembles a conducting state, its pore structure suggests that permeating anions have to shed their hydration shell and interact with pore-lining residues (Figure 6).”

*Furthermore, it is not clear to me that the "pore" of the cryo structure is large enough to accommodate a Cl ion, not to mention larger ions like iodide (radius 2.2Å) that is more permeant than Cl. It seems that there is barely enough space for a Cl ion to move through the pore of the Ala model. At the level of residues 546, 595, and 641 there is a constriction that is barely large enough to accommodate a Cl ion. With the significantly larger sidechains of the real amino acids at this location: N546, P595, and I641, there does not seem to be enough space for a Cl ion to pass. To get some insight into this, I replaced the Ala residues with the TMEM16A amino acids using the most common rotamer for each position and then energy minimized the structure. I understand that this structure could be completely bogus, but the aperture is >1Å too small in places to accommodate a Cl ion. This suggests that either this is a non-conducting state or that the conduction pathway is not located in the cavity between helices 4-6. I appreciate that the cryo structure may not have the resolution to allow the conclusions that I am suggesting, but yet the possibility that this is a non-conducting structure seems a possibility that must be addressed.*

Since the current structure of TMEM16A is based on an approximate poly-alanine model of the protein, we have intentionally avoided a quantitative analysis of the pore geometry, which requires a structure at high resolution. Consequently, we restricted our interpretation on the overall shape of the pore and its impact on ion conduction. The approximation of the pore shown in Figure 4—figure supplement 1, Figure 5 and Figure 5—figure supplement 2 results from a structure that was based on the refined poly-alanine coordinates where the positions of sidechains were substituted and subjected to few cycles of refinement as described in subsection “Electropysiology”. It was thus not calculated from the poly-alanine model. We have included this model in the resubmission of our manuscript for the reviewers to judge. The analysis with the program HOLE suggest a radius of the narrow part between 1.8 and 2.2 Å (see Figure 10). Figure 6 shows this pore in relation to modeled Cl^-^ ions (radius 1.8 Å) to illustrate that its general features are reasonable.

Author response image 1.Analysis of the pore radios along the channel axis as calculated by the program HOLE from a model of the channel containing side-chains.**DOI:**
http://dx.doi.org/10.7554/eLife.26232.026

Still, we want to emphasize that in its details, the model is likely not correct since, neither the location of the side-chains, nor the exact position of the main-chain atoms and their register are unambiguously defined. A detailed description of the pore will thus have to be provided with a high resolution structure.

To make this clear in our manuscript we have introduced the following changes:

Subsection “Functional properties of pore-mutations”:

“Since the current resolution of the data does not permit a quantitative analysis of its geometry, we restrict our description of the pore to its general geometric features. The wide, intracellular entrance narrows above the region constituting the regulatory Ca^2+^-binding site (Figure 4—figure supplement 1). Under the assumption that the structure is close to a conducting state, the narrow upper part most likely requires permeating ions to shed their hydration shell.”

*2) The mutagenesis experiments tend to support the idea that the Cl conduction pathway is the in the furrow (now a pore), but the residues chosen for mutation are located at the extreme ends of the enclosed pore (the α carbons of R515 and K588 are ~20 Å apart). While most of the residues that intervene between the vestibules are hydrophobic and perhaps not ideal candidates for mutagenesis analysis, it would be nice to have data on some amino acids that are located in the middle of the presumptive conduction pathway. The effect of the R515A mutation in particular is relatively small. Given the timeframe for revised submission, if additional experiments are not possible, the authors should discuss the limitations of their existing data.*

As we do not think that the current structure allows a detailed interpretation of the pore, we restricted ourselves to the investigation of the role of positively charged residues for anion conduction. For that purpose we have mutated all relevant basic amino acids in the pore region, most of which are located at both ends of the narrow neck and the intracellular vestibule. Due to the long-range nature of coulombic interactions and the low dielectric environment of the membrane the effect of the removal of positive charges is not restricted to their immediate surrounding. This is underlined by the strong effect of alanine mutations on conduction. Except for Lys 603 in the center of the neck, for which we were not able to record currents, most mutants were expressed on the surface of HEK cells and showed robust current response. Among these, the mutation of several residues showed a behavior that is consistent with their location along the pore lining. We strongly disagree that the effect of R515A is small. Given its robust expression, the mutation of this residue had a strong impact on the total currents and thus likely on the conductance of the channel. It should be noted that Figure 5 shows normalized currents, while the R515A current is six-fold lower in amplitude than for WT (see Figure 5—figure supplement 1). This is consistent with a predicted role for a mutant introducing a rate-limiting barrier for ion conduction in a model channel shown in Figure 5—figure supplement 2 and a fit to the data shown in Figure 5—figure supplement 2. It should also be emphasized that, due to the absence of defined loops at both ends, the register of the weakly conserved transmembrane α-helix 3, where R515 is located, is poorly defined. Its location might thus turn out to be deeper in the pore than anticipated from the current model. A detailed analysis of uncharged pore residues is beyond the scope of the current manuscript and will be subject of future investigations.

We have introduced the following changes to the manuscript:

In subsection “Functional properties of pore-mutations”:

“Due to the absence of a detailed structural representation of the ion conduction path, we focused on the role of long-range coulombic interactions on anion conduction. We have thus mutated basic residues in the pore to alanine (Figure 4—figure supplement 1) and recorded currents in inside-out patches (Figure 5—figure supplement 1).”

And

“Thus, this positive charge most likely lowers a rate-limiting energy barrier for anion permeation halfway through the narrow part of the mTMEM16A pore (Figure 5—figure supplement 2). This is consistent with the six-fold lower currents measured for this mutant, despite its robust expression at the surface of HEK cells (Figure 5—figure supplement 1).”

We have also introduced panel Figure 5—figure supplement 1 showing the fluorescence of the YFP-fusion construct of mutant R515A.

*3) The obtained resolution of 6.6Å from 4,000 K2-recorded micrographs is somewhat disappointing, and raises the question why no higher resolution was obtained. The 2D class averages look somewhat suboptimal. Still, the detailed data acquisition and image processing procedures described all appear to be performed to a high standard. Perhaps the protein sample itself wasn't optimal? Also, perhaps the sample micrograph in Figure 1—figure supplement 1 looks very/too contrasted. Could it be that the area imaged was too thin and the sample had dried too much? Anyway, these are not questions for the authors to answer now, but suggestion to consider for a next structure.*

*It is refreshing to see that the authors are well aware of its limitations, and are adequately careful in its interpretation. From the cryo-EM point-of-view, this paper is thereby sound and there are no technical reasons to reject it.*

We agree that in light of the large amount of data collected on a Titan Krios equipped with a K2 camera, the obtained resolution is disappointing. This is likely not due to deficits in the data collection since the alignment of the microscope was carefully checked and optimized during data collection. The quality of the 2D classes reflects in our opinion the limited resolution of the data and might be a consequence of the small size of the protein in a large digitonin micelle. The high contrast seen in the representative micrograph in Figure 1 is the result of the dose-weighted sum of 100 frames with a high total electron dose of 80 e^-^/Å (2). By comparing the electron dose of the incident beam with the transmitted beam, we assume that the ice might rather have been too thick than too thin. More importantly, as the addition of further images did not improve the resolution significantly, we believe that the bottleneck resides in the quality of the protein sample, which we are currently optimizing.

*4) Discussion section: "[…] as suggested by the observed distortion of the detergent micelle in mTMEM16A". The distortion is not at all clear from the figure. A better figure is needed to support this part of the discussion.*

We have provided an improved figure, see Figure 1—figure supplement 4.

*5) Subsection “Image processing”: "Overfitting was further prevented by correcting the final FSC curve for the effect of a soft mask using high-resolution noise substitution" This is not correct. The high-resolution noise substitution at this point is not to prevent overfitting, but rather to correct for convolution effects of real-space masking on the FSC curve.*

The description in Material & Methods was corrected:

“During post-processing, the approach of high-resolution noise substitution was used to correct for convolution effects of real-space masking on the FSC curve.”